# Association between the rs1544410 polymorphism in the vitamin D receptor (VDR) gene and insulin secretion after gestational diabetes mellitus

Nael Shaat[1,2]*, Anastasia Katsarou[1,2], Bushra Shahida[1], Rashmi B. Prasad[1], Karl Kristensen[1,3], Tereza Planck[1,2]

1 Department of Clinical Sciences, Genomics, Diabetes and Endocrinology, Lund University, Malmö, Sweden, 2 Department of Endocrinology, Skåne University Hospital, Malmö, Sweden, 3 Department of Obstetrics and Gynaecology, Skåne University Hospital, Malmö, Sweden

* nael.shaat@med.lu.se

## Abstract

### Background and aims

Genetic variants involved in vitamin D metabolism have been associated with diabetes and related syndromes/diseases. We wanted to investigate possible associations of polymorphisms in genes involved in vitamin D metabolism with indices of insulin resistance and insulin secretion, and also with development of diabetes after gestational diabetes mellitus (GDM).

### Materials and methods

We have studied 376 women with previous GDM. Eight single nucleotide polymorphisms (SNPs) in the genes for vitamin D receptor (*VDR*) [rs731236, rs7975232, rs10735810, and rs1544410], vitamin D binding protein (*DBP*) [rs7041 and rs4588], and cytochrome P450 family 27 subfamily B member 1 (*CYP27B1*) [rs10877012 and rs4646536] were genotyped by TaqMan Allelic Discrimination Assay using the Quantstudio 7 Flex system. A 75-g oral glucose tolerance test (OGTT) was performed 1–2 years postpartum. The homeostasis model assessment of insulin resistance (HOMA-IR) and the disposition index [(insulinogenic index: I30/G30)/HOMA-IR] were used to calculate insulin resistance and insulin secretion, respectively. Serum samples for determination of 25(OH)D3 were collected at the time of the OGTT. Manifestation of diabetes was followed up to five years postpartum.

### Results

After adjustment for BMI, age, and ethnicity, the A-allele of the *VDR* rs1544410 polymorphism was found to be associated with increased disposition index (difference per allele = 3.56, 95% CI: 0.4567–6.674; p = 0.03). The A-allele of the *DBP* rs7041 polymorphism was found to be associated with 25(OH)D3 levels (difference [in nmol/L] per allele = −5.478, 95% CI: -8.315 to −2.641; p = 0.0002), as was the T-allele of the *DBP* rs4588 polymorphism (OR

**Data Availability Statement:** All relevant data are within the manuscript and its Supporting Information files. EGA accession numbers are:

RNAseq: EGAS00001004042 GWAS: EGAS00001004044 Phenotype: EGAS00001004056.

**Funding:** The study was supported by grants from the Research Funds of Skåne University Hospital, the Skåne County Council Research and Development Foundation and ALF Region Skåne. The funders had no role in study design, data collection and analysis, decision to publish, or preparation of the manuscript.

**Competing interests:** The authors have declared that no competing interests exist.

**Abbreviations:** 25(OH)D3, 25-hydroxyvitamin D3; BMI, body mass index; CI, confidence interval; *CYP27B1*, cytochrome P450 family 27 subfamily B member 1; *DBP*, vitamin D binding protein; GDM, gestational diabetes mellitus; HOMA-IR, homeostasis model assessment of insulin resistance; OGTT, oral glucose tolerance test; OR, odds ratio; SNP, single nucleotide polymorphism; *VDR*, vitamin D receptor gene.

= −6.319, 95% CI: −9.466 to −3.171; p = 0.0001). None of the SNPs were significantly associated with HOMA-IR or postpartum diabetes.

## Conclusions

This study provides evidence that the rs1544410 polymorphism of the *VDR* gene may be associated with increased insulin secretion in women after pregnancy complicated by GDM. Further studies in other populations are needed to confirm the results.

## Introduction

Gestational diabetes mellitus (GDM) is defined as any degree of glucose intolerance with onset or first recognition during pregnancy [1]. The prevalence of GDM is increasing [2], in parallel with the increasing prevalence of type-2 diabetes (T2D) worldwide [3]. Women with a history of GDM have an increased risk of developing T2D relative to those with a normoglycemic pregnancy [4]. GDM is characterized by insulin resistance and beta-cell dysfunction, and the same is true for T2D [5].

Vitamin D has been implicated in several medical conditions [6]. It has been hypothesized that it may have a positive effect on insulin secretion and sensitivity. The mechanisms may be mediated by activation of the vitamin D receptor (VDR) on pancreatic beta cells and insulin-sensitive organs and also by regulation of calcium homeostasis [7, 8]. Studies have suggested that circulating concentrations of vitamin D may be inversely associated with the risk of diabetes, metabolic syndrome, insulin secretion, and insulin resistance [8–10]. Malik et al., in addition to a meta-analysis, have found associations between polymorphisms in the *VDR* gene and both diabetes and insulin resistance-related diseases [11, 12]. Moreover, single nucleotide polymorphisms (SNPs) in the vitamin D binding protein gene (*DBP*) affect 25(OH)D3 levels [13, 14] and increase the risk of T2D and the metabolic syndrome [15, 16]. There is also evidence of associations between polymorphisms in the *CYP27B1* gene and type-1 diabetes [17, 18].

We have recently reported from the "Mamma Study" that vitamin D deficiency/insufficiency in women with a history of GDM appeared to be associated with beta-cell dysfunction and insulin resistance, but not with postpartum diabetes, after adjustment for factors that are well known to influence T2D [19]. We therefore hypothesized that 8 SNPs in genes known to be involved in the vitamin D metabolism, diabetes, and related traits (4 SNPs in *VDR*, 2 SNPs in *DBP*, and 2 SNPs in *CYP27B1*) [8–18] might be associated with insulin resistance and/or beta-cell function at 1–2 years postpartum and also development of diabetes after up to 5 years of follow-up in women with a history of GDM.

## Material and methods

### Patients

The design of the "Mamma Study" has been described in detail elsewhere [20]. Briefly, women delivering between 2003–2005 were invited to participate in the study, covering 86% of all pregnancies in the County of Skane in southern Sweden, including four of five delivery departments. The diagnosis of GDM was made using a 75-g oral glucose tolerance test (OGTT) at the twenty-eighth and/or the twelfth week of gestation. In the original evaluation, GDM was defined as two-hour capillary plasma glucose ≥10.0 mmol/l, gestational IGT as two-hour

capillary plasma glucose 8.6–9.9 mmol/l, and normal glucose tolerance (NGT) during pregnancy as two-hour capillary plasma glucose <8.6 mmol/l. Among those who accepted to participate in the study and after exclusion of those who had already been diagnosed as having diabetes, 160 women had GDM, 309 had Gestational IGT and 167 had NGT. Women were followed for the development of diabetes using an OGTT at 1–2 years and 5 years after pregnancy, or until the diagnosis of diabetes. The World Health Organization (WHO) 1999 diagnostic criteria were used for classification of diabetes during follow-up [21]. Measurements of both glucose and insulin concentrations at 0, 30, and 120 min during OGTT at 1–2 years postpartum were performed to calculate indices of beta-cell function and insulin resistance, as previously reported [22]. Serum samples for determination of 25(OH)D3 were collected during the OGTT at 1–2 years postpartum. In the present investigation, we used the diagnostic criteria for GDM recommended by the WHO in 1999 [21]. Based on these criteria and on successful measurements of 25OHD3 concentrations, we identified 376 women who had previously had GDM and who formed the basis of the present study. All women gave their written informed consent and the Ethics Committee of Lund University approved the study (LU 259–00, 2002-04-17), which was performed in accordance with the Declaration of Helsinki.

## Chemical analysis

Homeostasis model assessment was used to estimate insulin resistance [(HOMA-IR): (fasting serum insulin × fasting plasma glucose)/22.5] [23, 24]. Insulin secretory capacity was estimated using the insulinogenic index (I/G30): [insulin $30\ \text{min}$−insulin $_{0\ \text{min}}$)/(glucose $_{30\ \text{min}}$ − glucose $_{0\ \text{min}}$] [25]. As insulin resistance modulates insulin secretion, the disposition index was used to adjust insulin secretion for the degree of insulin resistance, obtained by dividing I/G30 by HOMA-IR [26]. The serum concentration of 25(OH)D3 (in nmol/L) was determined by liquid chromatography mass spectrophotometry (LC-MS/MS). Assays were performed according to accredited methods at the Department of Clinical Chemistry, Skåne University Hospital, Malmö, Sweden.

## Genetic analysis

The SNPs were chosen for analysis based on previous associations with circulating vitamin D levels, diabetes, and related traits [8–18]. The following SNPs were genotyped: rs731236 (TaqI), rs7975232 (ApaI), rs10735810 (FokI), and rs1544410 (BsmI) in *VDR*; rs7041 and rs4588 in *DBP*; and rs10877012 and rs4646536 in *CYP27B1*. DNA was extracted from whole blood using the MaxiPrep Kit (QIAGEN, Sweden). SNPs were genotyped by TaqMan Allelic Discrimination Assay using the Quantstudio 7 Flex system. The genotyping results were confirmed by using positive controls during the genotyping as well as by re-genotyping of about 20% of the samples using the same genotyping method. The minor allele frequency for all SNPs was > 0.05. No deviation from Hardy-Weinberg equilibrium in the whole group or in the any of the subgroups was found. The success rate of genotyping was ≥96% (2 samples were not genotyped because of no available DNA). The genotyping protocol has been deposited in protocols.io. The DOI is: dx.doi.org/10.17504/protocols.io.bcjbiuin.

## eQTL lookups

The association of the rs1544410 polymorphism with gene expression in human pancreatic islets was looked up in data from RNAseq data from 191 donors [https://www.biorxiv.org/content/10.1101/435743v2.full]. The data are uploaded with the following EGA accession numbers: RNAseq: EGAS00001004042, GWAS: EGAS00001004044 and Phenotype: EGAS00001004056.

## Statistical analysis

All statistical analyses were performed using IBM SPSS Statistics for Windows, version 22.0 (IBM Corp., Armonk, NY). All genetic analyses were performed using PLINK version 1.07 (http://pngu.mgh.harvard.edu/~purcell/plink/index.shtml). Logistic and linear regressions with age, ethnicity, and body mass index (BMI) as covariates were used to estimate SNP associations and the data are presented as difference per allele or odds ratios (ORs) with 95% confidence intervals (CIs). The p-values are based on additive models for the genetic variants. Correction for multiple testing was performed using permutations.

## Results

The clinical characteristics of the study subjects are presented in a previous publication [19]. The study included 376 women who were diagnosed with GDM during pregnancy in southern Sweden. At five years postpartum, 253 (67.3%) had normal glucose tolerance, 57 (15.2%) had diabetes, and for 66 (17.6%) there were missing data. The women were of European origin (n = 287, 76.3%) or non-European origin (n = 78, 20.8%) but 11 women (2.9%) with GDM were unclassifiable. The non-European group consisted of the subgroups Arab women (n = 33), Asian women (n = 35), and women of other origins (n = 10).

Of the SNPs tested, the A-allele of the rs1544410 polymorphism was associated with increased disposition index (difference per allele = 3.56, CI: 0.46–6.67; p = 0.03) using linear regression analysis with age, ethnicity, and BMI as covariates (Table 1).

The rs731236 (Taql), rs7975232 (Apal), and rs1544410 (Bsml) polymorphisms of the *VDR* gene were in complete linkage disequilibrium (Fig 1), as previously shown [27]. However, the rs10735810 (FokI) polymorphism was not in linkage disequilibrium with the SNPs mentioned.

However, we found no effect of the SNPs in question on the HOMA-IR, using linear multiple regression analysis (Table 2).

In logistic regression analysis with age, ethnicity, and BMI as covariates, the SNPs studied showed no associations with postpartum diabetes in women with a history of GDM (Table 3).

The A-allele of the rs7041 polymorphism of the *DBP* gene was associated with a reduction in circulating 25(OH)D3 (difference [in nmol/L] per allele = −5.48, 95% CI: −8.32 to −2.64; p = 0.0002). The same was observed for the T-allele of the rs4588 polymorphism (difference [in nmol/L] per allele = −6.32, 95% CI: −9.47 to −3.17; p = 0.0001). The other SNPs did not show any effect on circulating levels of 25(OH)D3 (Table 4).

The rs1544410 polymorphism was an eQTL for the KAT8 regulatory NSL complex subunit 2 (*KANSL2*) and phosphofructokinase, muscle (*PFKM*) genes in human pancreatic islets

**Table 1. The association between the SNPs studied and disposition index.**

| SNP | Associated allele | Frequency of associated allele | Difference in disposition index per allele (95% CI) | p-value |
|---|---|---|---|---|
| rs7041 | A | 0.41 | −0.47 (−2.71 to 1.76) | 0.68 |
| rs4588 | T | 0.38 | −0.62 (−3.14 to 1.91) | 0.63 |
| rs731236 | G | 0.38 | 0.25 (−1.90 to 2.40) | 0.82 |
| rs7975232 | C | 0.40 | −1.19 (−3.31 to 0.92) | 0.27 |
| rs10735810 | A | 0.38 | 0.62 (−1.59 to 2.82) | 0.58 |
| rs1544410 | A | 0.12 | 3.56 (0.46 to 6.67) | **0.03** |
| rs10877012 | T | 0.34 | 0.97 (−1.36 to 3.30) | 0.42 |
| rs4646536 | G | 0.34 | 1.05 (−1.30 to 3.37) | 0.38 |

The associations were tested by using linear regression analysis with age, ethnicity, and BMI as covariates.

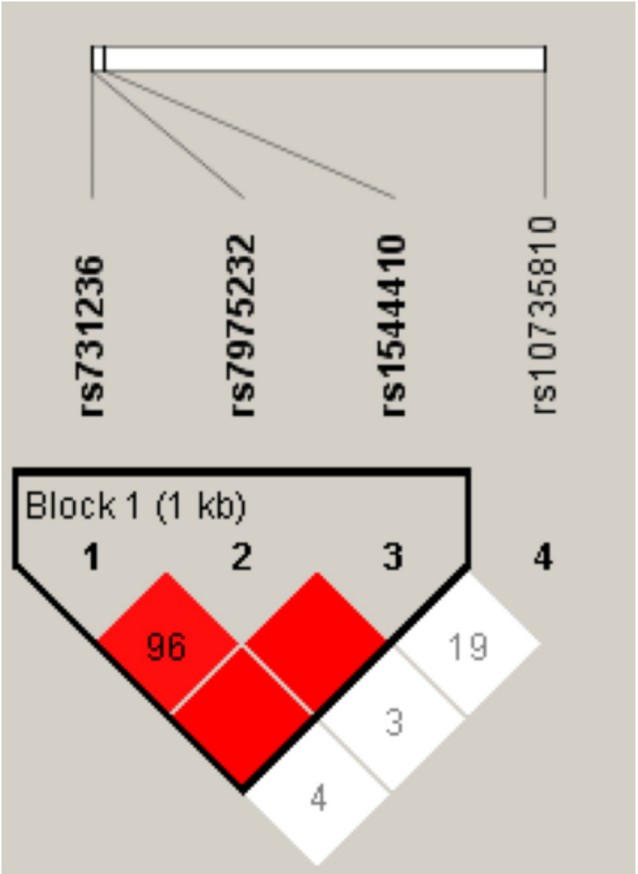

**Fig 1. Linkage disequilibrium between the rs731236, rs7975232, rs10735810, and rs1544410 polymorphisms of the *VDR* gene.**

[https://www.biorxiv.org/content/10.1101/435743v2.full] (Table 5). *KANSL2* gene expression correlated with that of Islet Amyloid Polypeptide (*IAPP*) gene (Fig 2). *PFKM* expression correlated with stimulatory index (a measure of insulin secretion in islets), as well as with expression of insulin, glucagon, and IAPP genes (Fig 3) [https://www.biorxiv.org/content/10.1101/435743v2.full].

**Table 2. The association between the SNPs studied and HOMA-IR.**

| SNP | Allele | Allele Frequency | Difference in HOMA-IR per allele (95% CI) | p-value |
|---|---|---|---|---|
| **rs7041** | A | 0.41 | 0.05 (−0.14 to 0.24) | 0.62 |
| **rs4588** | T | 0.38 | 0.01 (−0.20 to 0.22) | 0.93 |
| **rs731236** | G | 0.38 | −0.02 (−0.20 to 0.16) | 0.83 |
| **rs7975232** | C | 0.40 | 0.01 (−0.16 to 0.18) | 0.91 |
| **rs10735810** | A | 0.38 | −0.07 (−0.26 to 0.12) | 0.45 |
| **rs1544410** | A | 0.12 | −0.07 (−0.34 to 0.20) | 0.60 |
| **rs10877012** | T | 0.34 | −0.05 (−0.25 to 0.15) | 0.61 |
| **rs4646536** | G | 0.34 | −0.004 (−0.20 to 0.20) | 0.97 |

The associations were tested by using linear regression analysis with age, ethnicity, and BMI as covariates.

**Table 3. The association between the SNPs studied and postpartum diabetes.**

| SNP | Associated allele | Frequency of associated allele | OR (95% CI) | p-value |
|-----|-----|-----|-----|-----|
| rs7041 | A | 0.41 | 1.01 (0.66–1.54) | 0.98 |
| rs4588 | G | 0.62 | 1.42 (0.89–2.26) | 0.14 |
| rs731236 | A | 0.62 | 1.42 (0.95–2.12) | 0.09 |
| rs7975232 | C | 0.40 | 1.15 (0.77–1.72) | 0.50 |
| rs10735810 | G | 0.62 | 1.16 (0.76–1.78) | 0.48 |
| rs1544410 | A | 0.12 | 1.64 (0.81–3.30) | 0.17 |
| rs10877012 | G | 0.66 | 1.12 (0.73–1.73) | 0.60 |
| rs4646536 | A | 0.66 | 1.10 (0.71–1.68) | 0.68 |

The associations were tested using logistic regression analysis with age, ethnicity, and BMI as covariates.

## Discussion

We have recently shown that vitamin D deficiency/insufficiency in women with a history of GDM is associated with beta-cell dysfunction and insulin resistance [19]. The key finding of the current study was that the A-allele of the rs1544410 polymorphism of the *VDR* gene was associated with increased insulin secretion, measured as disposition index in women with a history of GDM. VDR is a member of the nuclear receptor superfamily of transcriptional regulators. It forms a heterodimer with a retinoid X receptor (RXR) and is widely distributed throughout the tissues, including pancreatic beta cells [28]. The 25(OH)D3 is widely used in assessment of vitamin D status. 25(OH)D3 is converted in the liver to 1,25(OH)2D, which is the main ligand for VDR [29]. When 1,25(OH)2D binds to this complex, it leads to the transcription of several vitamin D-regulated genes [23]. A relatively recent study has shown increased expression and production of major renin-angiotensin system components in isolated islets from VDR knockout mice incubated under high-glucose conditions [30]. This was prevented and reversed by 1,25(OH)2D in parallel with an increase in glucose-stimulated insulin secretion [30]. Ogunkolade et al. have shown that the TaqI (rs731236) genotype of the *VDR* gene was an independent determinant of the insulin secretion index in healthy Bangladeshi adults, who have a high risk of developing T2D [27]. The FokI (rs10735810) polymorphism of the *VDR* gene was found to be an additional independent determinant of insulin secretion in individuals with vitamin D insufficiency [27]. Furthermore, *VDR* mRNA expression has been shown to be a determinant of insulin secretory capacity [27]. Unlike our study, there was no

**Table 4. The association between the SNPs tested and circulating 25(OH)D3.**

| SNP | Associated allele | Frequency of associated allele | Difference in 25(OH)D3 (in nmol/L) per allele (95% CI) | p-value |
|-----|-----|-----|-----|-----|
| rs7041 | A | 0.41 | −5.48 (−8.32 to −2.64) | **0.0002** |
| rs4588 | T | 0.38 | −6.32 (−9.47 to −3.17) | **0.0001** |
| rs731236 | G | 0.38 | 0.20 (−2.57 to −2.97) | 0.89 |
| rs7975232 | C | 0.40 | 1.01 (−1.71 to 3.73) | 0.47 |
| rs10735810 | A | 0.38 | −0.87 (−3.78 to 2.05) | 0.56 |
| rs1544410 | A | 0.12 | −0.66 (−4.75 to 3.42) | 0.75 |
| rs10877012 | T | 0.34 | −1.16 (−4.14 to 1.82) | 0.45 |
| rs4646536 | G | 0.34 | −1.14 (−4.08 to 1.81) | 0.45 |

The associations were tested using linear regression analysis with age, ethnicity, and BMI as covariates

**Table 5. The rs1544410 polymorphism was an eQTL for the *KANSL2* and *PFKM* genes in human pancreatic islets.**

| SNP | SNP_rsid | Gene name | beta | t-stat | p-value |
|---|---|---|---|---|---|
| 12:48239835 | rs1544410 | KANSL2 | 0,29472031 | 2,864151456 | 0,004661904 |
| 12:48239835 | rs1544410 | PFKM | 0,223429837 | 2,167193122 | 0,031491255 |

"Lookup" of significant associations in human pancreatic islets from 191 donors

direct association of the BsmI (rs1544410) polymorphism with insulin secretion. Interestingly, the TaqI t allele (i.e. the G allele in the present study) and the BsmI b (G) allele may be susceptibility alleles for diabetes in the Kashmiri population [11]. This is consistent with our present finding that the opposite allele, the A-allele, of the rs1544410 polymorphism was associated with increased disposition index, and therefore would prevent the development of diabetes. On the other hand, the TaqI t homozygotes were found to be associated with the highest levels of insulin secretion index compared to TT and Tt genotypes in healthy Bangladeshi adults, who have a high risk of developing T2D [27]. However, it should be noted that there was no adjustment for the degree of insulin resistance (i.e. disposition index), as we have done in the

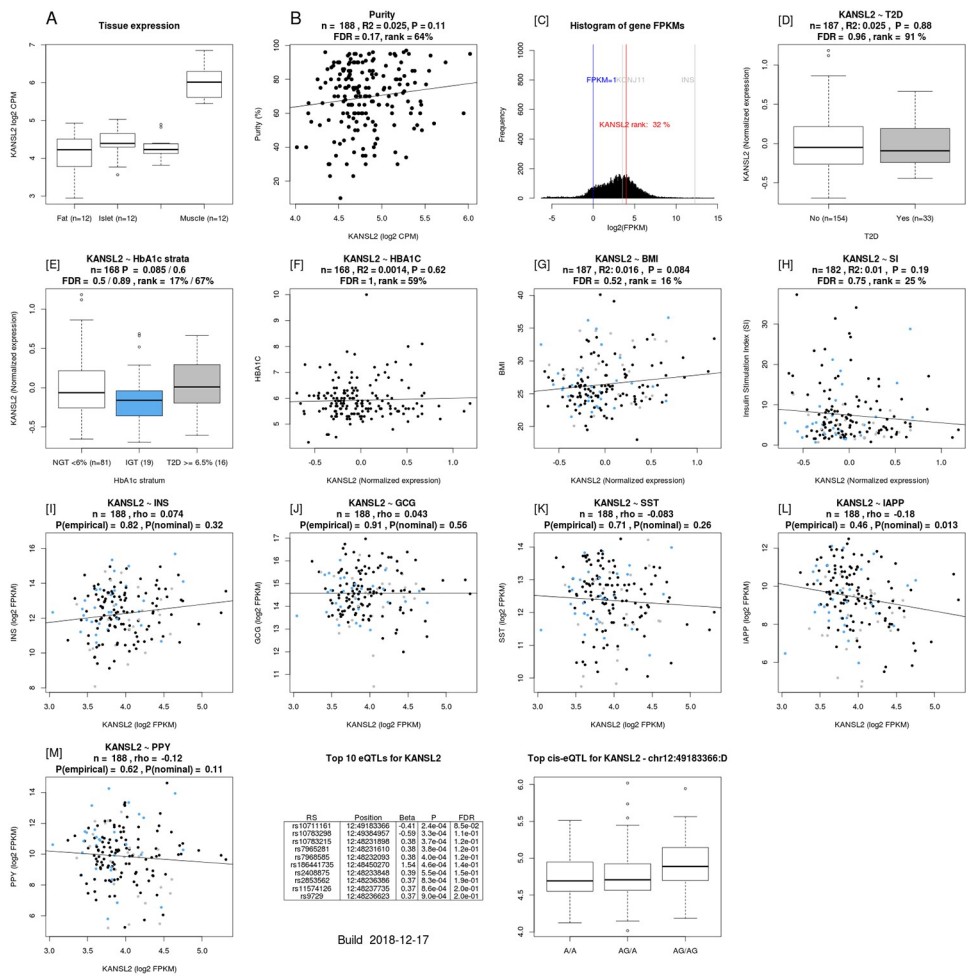

**Fig 2. *KANSL2* gene expression is correlated with that of *IAPP* gene.**

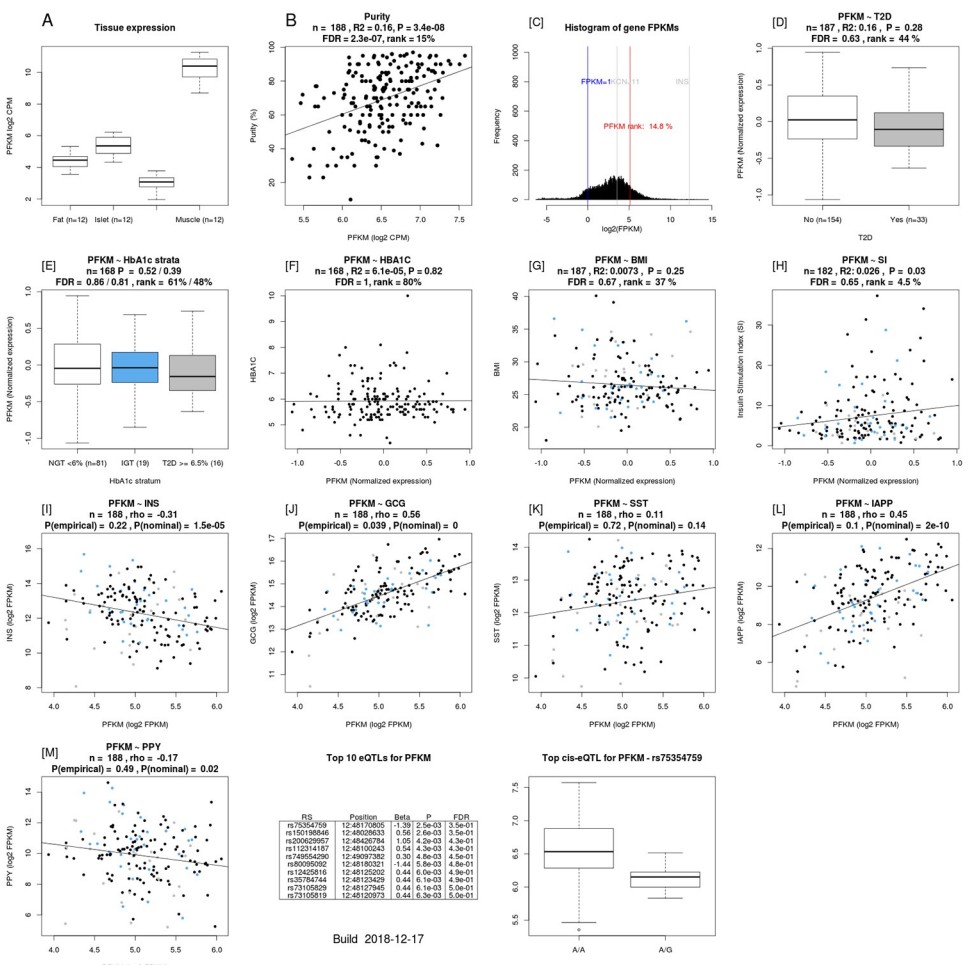

**Fig 3. The correlation of the *PFKM* expression with stimulatory index as well as with expression of *insulin*, *glucagon*, and *IAPP* genes.**

present study. Another explanation might be that the susceptibility alleles differ in different populations. Previous studies have shown tight linkage disequilibrium between the ApaI, BsmI, and TaqI genotypes [27, 31], and the same was found in our study (Fig 1). Our finding that the rs1544410 polymorphism was an eQTL for the *PFKM* gene in human pancreatic islets (Table 4) and that the *PFKM* expression correlated with stimulatory index (a measure of insulin secretion in islets), as well as with expression of *INS*, *GCG*, and *IAPP* genes (Fig 3) supports the role of this SNP in the insulin secretory capacity.

In the Rancho Bernardo study, the BsmI bb (GG) genotype was found to be associated with HOMA-IR after adjustment for age, sex, BMI, calcium, and vitamin D use in older adults without diabetes [32]. In the present study, we found no association between the polymorphisms tested and the HOMA-IR. Interestingly, a recent meta-analysis provided evidence of an association between *VDR* BsmI polymorphism and metabolic syndrome, supporting the idea that the *VDR* BsmI variant G allele may be a susceptibility marker of metabolic syndrome [12]. On the other hand, gene carriers with the *VDR* BsmI BB (AA) genotype have been found to have significantly higher levels of fasting glucose than gene carriers with the genotype Bb or bb in young men from Germany with low physical activity, but the effect was absent in men with a

high degree of physical activity [33]. This discrepancy in the risk allele perhaps represents a gender difference.

In addition, we found that the A-allele of the rs7041 polymorphism and the T-allele of the rs4588 polymorphism of the *DBP* gene were associated with lower levels of circulating 25(OH) D3. The composite genotype of these 2 SNPs (rs7041 and rs4588) results in different DBP isotypes (Gc1f, Gc1s, and Gc2). These isotypes are known to influence the binding affinity of DBP and circulating vitamin D levels [34, 35]. Our results are consistent with previous findings that T-allele carriers (i.e. A-allele carriers in this study) of the rs7041 polymorphism had less circulating 25(OH)D3, regardless of reproductive state among third-trimester pregnant, lactating, and non-pregnant/non-lactating women [13]. Moreover, the TT genotype of the rs7041 polymorphism was found to be associated with metabolic syndrome in PCOS women, and with significantly lower 25(OH)D3 levels in both PCOS and control groups [15]. This was also shown in Jordanian subjects, where the genotypes containing the variant allele of rs7041 (TT, TG) and rs4588 (AA, AC) (i.e. T-allele in this study) were found to be associated with lower 25 (OH)D3 levels than the wild-type genotypes (GG and CC, respectively) [14]. Interestingly, Rahman et al. found that the Glu/Glu genotype of the rs7041 and the Lys/Lys genotype of the rs4588 variants of *DBP* gene were significantly higher in subjects with type 2 diabetes than controls [16].

We have not observed any association between the SNPs studied and postpartum diabetes in the present study, which could be due to the small number of women who developed diabetes, to differences in ethnic background, to differences in study design, or simply to absence of an effect of these SNPs on the development of postpartum diabetes.

The main strength of the study was that we used the disposition index derived from the OGTT as a measure of beta-cell function, adjusted for insulin resistance. This is a more accurate measure of insulin secretory capacity than the HOMA-b [25]. However, the study still lacked enough statistical power (i.e. > 80%) to detect small potential effects of the SNPs in question on prediction of postpartum diabetes.

In conclusion, our study provides evidence that the rs1544410 polymorphism of the *VDR* gene may be associated with insulin secretion. Further studies in other populations will be needed to confirm the results.

## Supporting information

**S1 Data.**
(SAV)

## Author Contributions

**Conceptualization:** Nael Shaat, Tereza Planck.

**Data curation:** Nael Shaat.

**Formal analysis:** Rashmi B. Prasad, Tereza Planck.

**Investigation:** Bushra Shahida, Rashmi B. Prasad.

**Supervision:** Tereza Planck.

**Writing – original draft:** Nael Shaat.

**Writing – review & editing:** Nael Shaat, Anastasia Katsarou, Bushra Shahida, Rashmi B. Prasad, Karl Kristensen, Tereza Planck.

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
