## [Decision Letter · Decision Letter 0]

2 Jan 2020

PONE-D-19-32028

Association between the rs1544410 polymorphism in the vitamin D receptor (VDR) gene and insulin secretion after gestational diabetes mellitus

PLOS ONE

Dear Dr. Shaat,

Thank you for submitting your manuscript to PLOS ONE. After careful consideration, we feel that it has merit but does not fully meet PLOS ONE’s publication criteria as it currently stands. Therefore, we invite you to submit a revised version of the manuscript that addresses the points raised during the review process.

ACADEMIC EDITOR: 

I request the authors to clear the following queries

1. Explain how the study size was arrived at.

2. Did the authors confirmed the genotyping results using any other methods (sequencing etc.)

3.Please include the genotyping raw data (spss file etc.) as supplementary information.

We would appreciate receiving your revised manuscript by Feb 16 2020 11:59PM. To enhance the reproducibility of your results, we recommend that if applicable you deposit your laboratory protocols in protocols.io, where a protocol can be assigned its own identifier (DOI) such that it can be cited independently in the future. For instructions see: http://journals.plos.org/plosone/s/submission-guidelines#loc-laboratory-protocols

We look forward to receiving your revised manuscript.

Kind regards,

Narasimha Reddy Parine, Ph.D

Academic Editor

PLOS ONE

Journal Requirements:

2. We note that you are reporting an analysis of a microarray, next-generation sequencing, or deep sequencing data set. PLOS requires that authors comply with field-specific standards for preparation, recording, and deposition of data in repositories appropriate to their field. Please upload these data to a stable, public repository (such as ArrayExpress, Gene Expression Omnibus (GEO), DNA Data Bank of Japan (DDBJ), NCBI GenBank, NCBI Sequence Read Archive, or EMBL Nucleotide Sequence Database (ENA)). In your revised cover letter, please provide the relevant accession numbers that may be used to access these data. For a full list of recommended repositories, see http://journals.plos.org/plosone/s/data-availability#loc-omics or http://journals.plos.org/plosone/s/data-availability#loc-sequencing.

3. Please note that PLOS does not permit references to “data/results not shown.” Authors should provide the relevant data within the manuscript, the Supporting Information files, or in a public repository. If the data are not a core part of the research study being presented, we ask that authors remove any references to these data.

"The study was supported by grants from the Research Funds of Skåne University Hospital and

289 the Skåne County Council Research and Development Foundation. The funders had no role in

290 study design, data collection and analysis, decision to publish, or preparation of the manuscript.".

 "The funders had no role in study design, data collection and analysis, decision to publish, or preparation of the manuscript.".

Additional Editor Comments (if provided):

I request the authors to clear the following queries

1. Explain how the study size was arrived at.

2. Did the authors confirmed the genotyping results using any other methods (sequencing etc.)

3.please include the genotyping raw data (spss file etc.) as supplementary information.

Reviewers' comments:

Reviewer's Responses to Questions

**Comments to the Author**

1. Is the manuscript technically sound, and do the data support the conclusions?

Reviewer #1: Yes

Reviewer #2: Yes

2. Has the statistical analysis been performed appropriately and rigorously? 

Reviewer #1: Yes

Reviewer #2: Yes

3. Have the authors made all data underlying the findings in their manuscript fully available?

Reviewer #1: Yes

Reviewer #2: Yes

4. Is the manuscript presented in an intelligible fashion and written in standard English?

Reviewer #1: Yes

Reviewer #2: Yes

5. Review Comments to the Author

Reviewer #1: The study aimed to investigate possible associations of polymorphisms in genes involved in vitamin D metabolism with indices of insulin resistance and insulin secretion, and also with development of diabetes after gestational diabetes mellitus (GDM), the manuscript was found interesting, The subject selection and how subjects move through the study is clear. It is a good work.

Reviewer #2: I would like to appreciate the work done by the authors submitted for publication. Overall the manuscript is well designed, experimental study is well executed, statistical data analysis performed is acceptable. Overall the manuscript is well written and the subject of study and findings reported are relevant.

6. PLOS authors have the option to publish the peer review history of their article (what does this mean?). If published, this will include your full peer review and any attached files.

Reviewer #1: Yes: Moushira Erfan Zaki

Reviewer #2: No

---

## [Author Response · Author response to Decision Letter 0]

9 Mar 2020

Response to Reviewers

Academic Editor: 

I request the authors to clear the following queries

1. Explain how the study size was arrived at.

Authors’ reply: Briefly, women delivering between 2003–2005 were invited to participate in the study, covering 86% of all pregnancies in the County of Skane in southern Sweden, including four of ﬁve delivery departments. The diagnosis of GDM was made using a 75-g oral glucose tolerance test (OGTT) at the twenty-eighth and/or the twelfth week of gestation. In the original evaluation, GDM was deﬁned as two-hour capillary plasma glucose ≥10.0 mmol/l, gestational IGT as two-hour capillary plasma glucose 8.6–9.9 mmol/l, and normal glucose tolerance (NGT) during pregnancy as two-hour capillary plasma glucose <8.6 mmol/l. Among those who accepted to participate in the study and after exclusion of those who had already been diagnosed as having diabetes, 160 women had GDM, 309 had Gestational IGT and 167 had NGT. Women were followed for the development of diabetes using an OGTT at 1–2 years and 5 years after pregnancy, or until the diagnosis of diabetes. The World Health Organization (WHO) 1999 diagnostic criteria were used for classification of diabetes during follow-up. Measurements of both glucose and insulin concentrations at 0, 30, and 120 min during OGTT at 1‒2 years postpartum were performed to calculate indices of beta-cell function and insulin resistance, as previously reported. Serum samples for determination of 25(OH)D3 were collected during the OGTT at 1‒2 years postpartum. In the present investigation, we used the diagnostic criteria for GDM recommended by the WHO in 1999. Based on these criteria and on successful measurements of 25OHD3 concentrations, we identified 376 women who had previously had GDM and who formed the basis of the present study. 

We have now revised the Section “Material and methods/Patients” accordingly. 

2. Did the authors confirmed the genotyping results using any other methods (sequencing etc.)

Authors’ reply: As the TaqMan Allelic Discrimination Assay is well validated method for SNP genotyping, we have not used any other methods to confirm the genotyping. However, we used positive controls during the genotyping. Moreover, the genotyping results were confirmed by re-genotyping of about 20% of the samples using the same genotyping method. 

We think this would be enough to confirm our results and hope it is accepted according to your journal policies. We have now added this information to the Section “Material and methods/Genetic analysis”. 

3. Please include the genotyping raw data (spss file etc.) as supplementary information.

Authors’ reply: We would like to bring to your attention that we discovered that two samples of the studied samples were not genotyped because of no available DNA. We have now added this information to the Section “Material and methods/Genetic analysis”. The genotyping raw data is included in SPSS file as supplementary information. Please, let us to know if you need more raw data (t. ex. original genotyping raw data from the TaqMan ® Genotyper ™ Software). 

• Authors’ reply: No changes to our financial disclosure have been made. 

• To enhance the reproducibility of your results, we recommend that if applicable you deposit your laboratory protocols in protocols.io, where a protocol can be assigned its own identifier (DOI) such that it can be cited independently in the future. For instructions see: http://journals.plos.org/plosone/s/submission-guidelines#loc-laboratory-protocols

Authors’ reply: The genotyping protocol have been deposited in protocols.io. The DOI is: dx.doi.org/10.17504/protocols.io.bcjbiuin

• A rebuttal letter that responds to each point raised by the academic editor and reviewer(s). This letter should be uploaded as separate file and labeled 'Response to Reviewers'.

Authors’ reply: This is done.

• A marked-up copy of your manuscript that highlights changes made to the original version. This file should be uploaded as separate file and labeled 'Revised Manuscript with Track Changes'.

Authors’ reply: This is done.

• An unmarked version of your revised paper without tracked changes. This file should be uploaded as separate file and labeled 'Manuscript'.

Authors’ reply: This is done.

Journal Requirements:

Authors’ reply: Your journal requirements have been taken into consideration during revision of the manuscript. 

2. We note that you are reporting an analysis of a microarray, next-generation sequencing, or deep sequencing data set. PLOS requires that authors comply with field-specific standards for preparation, recording, and deposition of data in repositories appropriate to their field. Please upload these data to a stable, public repository (such as ArrayExpress, Gene Expression Omnibus (GEO), DNA Data Bank of Japan (DDBJ), NCBI GenBank, NCBI Sequence Read Archive, or EMBL Nucleotide Sequence Database (ENA)). In your revised cover letter, please provide the relevant accession numbers that may be used to access these data. For a full list of recommended repositories, see http://journals.plos.org/plosone/s/data-availability#loc-omics or http://journals.plos.org/plosone/s/data-availability#loc-sequencing.

Authors’ reply: The data are uploaded. The EGA accession numbers are:

RNAseq: EGAS00001004042

GWAS: EGAS00001004044

Phenotype: EGAS00001004056

This information has been added to the revised manuscript. 

3. Please note that PLOS does not permit references to “data/results not shown.” Authors should provide the relevant data within the manuscript, the Supporting Information files, or in a public repository. If the data are not a core part of the research study being presented, we ask that authors remove any references to these data.

Authors’ reply: We have now provided data within the manuscript. Please, see “Table 2” in the Results Section.

"The study was supported by grants from the Research Funds of Skåne University Hospital and

289 the Skåne County Council Research and Development Foundation. The funders had no role in

290 study design, data collection and analysis, decision to publish, or preparation of the manuscript.".

Authors’ reply: We have got a little bit confused. You request in the next point that funding information should not appear in the Acknowledgments section or other areas of our manuscript. Please let us know if we have misunderstood your requirements. Please, see our response to the next point.

 "The funders had no role in study design, data collection and analysis, decision to publish, or preparation of the manuscript.".

Authors’ reply: We have removed any funding-related text from the manuscript. We want to update our funding information as follow: “The study was supported by grants from the Research Funds of Skåne University Hospital, the Skåne County Council Research and Development Foundation and ALF Region Skåne. The funders had no role in study design, data collection and analysis, decision to publish, or preparation of the manuscript.”

Reviewers' comments:

Reviewer's Responses to Questions

Comments to the Author

1. Is the manuscript technically sound, and do the data support the conclusions?

Reviewer #1: Yes

Reviewer #2: Yes

 Authors’ reply: There are no questions raised by the reviewer.

2. Has the statistical analysis been performed appropriately and rigorously? 

Reviewer #1: Yes

Reviewer #2: Yes

 Authors’ reply: There are no questions raised by the reviewer.

3. Have the authors made all data underlying the findings in their manuscript fully available?

Reviewer #1: Yes

Reviewer #2: Yes

 Authors’ reply: There are no questions raised by the reviewer.

4. Is the manuscript presented in an intelligible fashion and written in standard English?

Reviewer #1: Yes

Reviewer #2: Yes

Authors’ reply: There are no questions raised by the reviewer.

5. Review Comments to the Author

Reviewer #1: The study aimed to investigate possible associations of polymorphisms in genes involved in vitamin D metabolism with indices of insulin resistance and insulin secretion, and also with development of diabetes after gestational diabetes mellitus (GDM), the manuscript was found interesting, The subject selection and how subjects move through the study is clear. It is a good work.

Authors’ reply: There are no questions raised by the reviewer.

Reviewer #2: I would like to appreciate the work done by the authors submitted for publication. Overall the manuscript is well designed, experimental study is well executed, statistical data analysis performed is acceptable. Overall the manuscript is well written and the subject of study and findings reported are relevant.

 Authors’ reply: There are no questions raised by the reviewer.

6. PLOS authors have the option to publish the peer review history of their article (what does this mean?). If published, this will include your full peer review and any attached files.

Do you want your identity to be public for this peer review? For information about this choice, including consent withdrawal, please see our Privacy Policy.

Reviewer #1: Yes: Moushira Erfan Zaki

Reviewer #2: No

---

## [Decision Letter · Decision Letter 1]

13 Apr 2020

Association between the rs1544410 polymorphism in the vitamin D receptor (VDR) gene and insulin secretion after gestational diabetes mellitus

PONE-D-19-32028R1

Dear Dr. Shaat,

We are pleased to inform you that your manuscript has been judged scientifically suitable for publication and will be formally accepted for publication once it complies with all outstanding technical requirements.

With kind regards,

Narasimha Reddy Parine, Ph.D

Academic Editor

PLOS ONE

Reviewers' comments:

Reviewer's Responses to Questions

**Comments to the Author**

1. If the authors have adequately addressed your comments raised in a previous round of review and you feel that this manuscript is now acceptable for publication, you may indicate that here to bypass the “Comments to the Author” section, enter your conflict of interest statement in the “Confidential to Editor” section, and submit your "Accept" recommendation.

Reviewer #1: All comments have been addressed

Reviewer #2: All comments have been addressed

2. Is the manuscript technically sound, and do the data support the conclusions?

Reviewer #1: Yes

Reviewer #2: Yes

3. Has the statistical analysis been performed appropriately and rigorously? 

Reviewer #1: Yes

Reviewer #2: Yes

4. Have the authors made all data underlying the findings in their manuscript fully available?

Reviewer #1: Yes

Reviewer #2: Yes

5. Is the manuscript presented in an intelligible fashion and written in standard English?

Reviewer #1: Yes

Reviewer #2: Yes

6. Review Comments to the Author

Reviewer #1: no comments for authors, good work, the conclusion showed that he overall early ANC booking in Ethiopia was very low compared with

WHO recommendation for pregnant women. Both sociodemographic and obstetric

factors significantly affect early initiation of ANC. Strategies targeting advocating

women education and standard piece of information concerning what pregnant women

should expect and do during the prenatal period, universal access of family planning

and prenatal care service and establishment village outreach health facility should be

addressed by Ministry of Health and its stake holders to improve early initiation of ANC.

Reviewer #2: (No Response)

7. PLOS authors have the option to publish the peer review history of their article (what does this mean?). If published, this will include your full peer review and any attached files.

Reviewer #1: Yes: Moushira Zaki, National Research Centre,Egypt

Reviewer #2: Yes: Prof. Dr. Nadeem Sheikh

---

## [Editor Report · Acceptance letter]

7 May 2020

PONE-D-19-32028R1 

Association between the rs1544410 polymorphism in the vitamin D receptor (VDR) gene and insulin secretion after gestational diabetes mellitus 

Dear Dr. Shaat:

I am pleased to inform you that your manuscript has been deemed suitable for publication in PLOS ONE. Congratulations! Your manuscript is now with our production department. 

With kind regards,

on behalf of

Dr. Narasimha Reddy Parine 

Academic Editor

PLOS ONE